# OmniSync: Towards Universal Lip Synchronization via Diffusion Transformers

**Ziqiao Peng**[1]* **Jiwen Liu**[2]† **Haoxian Zhang**[2] **Xiaoqiang Liu**[2] **Songlin Tang**[2]
**Pengfei Wan**[2] **Di Zhang**[2] **Hongyan Liu**[3]✉ **Jun He**[1]✉
[1]Renmin University of China [2]Kling Team, Kuaishou Technology [3]Tsinghua University
†Project Leader ✉Corresponding Author

https://ziqiaopeng.github.io/OmniSync/

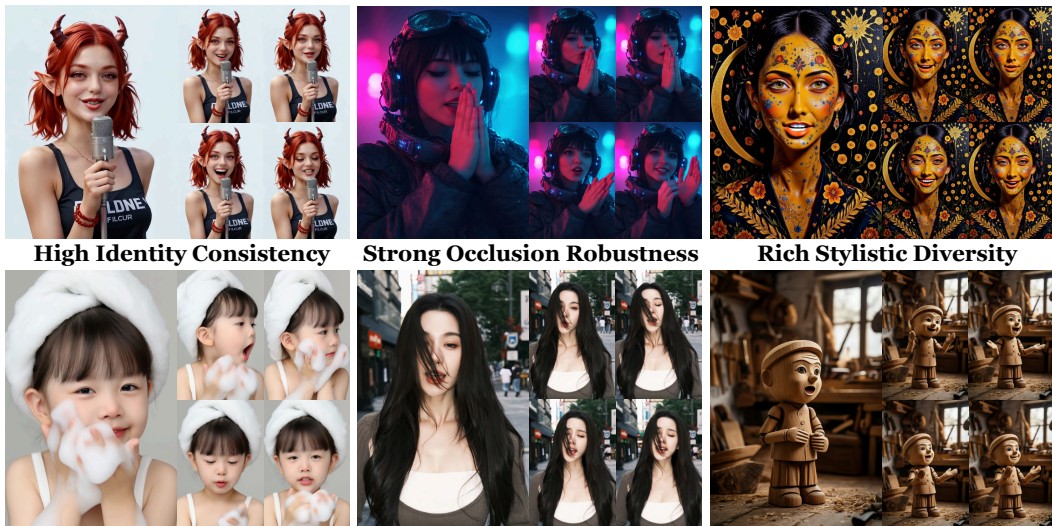

**High Identity Consistency** **Strong Occlusion Robustness** **Rich Stylistic Diversity**

Figure 1: **OmniSync** demonstrates universal lip synchronization capabilities, effectively handling facial occlusion, while maintaining visual consistency and generating accurate lip movements.

## Abstract

Lip synchronization is the task of aligning a speaker's lip movements in video with corresponding speech audio, and it is essential for creating realistic, expressive video content. However, existing methods often rely on reference frames and masked-frame inpainting, which limit their robustness to identity consistency, pose variations, facial occlusions, and stylized content. In addition, since audio signals provide weaker conditioning than visual cues, lip shape leakage from the original video will affect lip sync quality. In this paper, we present OmniSync, a universal lip synchronization framework for diverse visual scenarios. Our approach introduces a mask-free training paradigm using Diffusion Transformer models for direct frame editing without explicit masks, enabling unlimited-duration inference while maintaining natural facial dynamics and preserving character identity. During inference, we propose a flow-matching-based progressive noise initialization to ensure pose and identity consistency, while allowing precise mouth-region editing. To address the weak conditioning signal of audio, we develop a Dynamic Spatiotemporal Classifier-Free Guidance (DS-CFG) mechanism that adaptively adjusts guidance strength over time and space. We also establish the AIGC-LipSync Benchmark, the first evaluation suite for lip synchronization in diverse AI-generated videos.

---

*Work done during an internship at Kling Team, Kuaishou Technology.

39th Conference on Neural Information Processing Systems (NeurIPS 2025).

Extensive experiments demonstrate that OmniSync significantly outperforms prior methods in both visual quality and lip sync accuracy, achieving superior results in both real-world and AI-generated videos.

# 1 Introduction

Lip synchronization, matching mouth movements with speech audio, is essential for creating compelling visual content across film dubbing [51], digital avatars [32, 31, 55, 5], and teleconferencing [24, 28, 53]. With the rise of AI-generated content, this technology has evolved from a specialized technique to a fundamental aspect of the video generation landscape [47, 34, 20]. Despite significant advances in text-to-video (T2V) models [4, 2, 48, 16, 39] creating increasingly realistic footage, achieving precise and natural lip synchronization remains an unsolved challenge.

Traditional lip synchronization approaches rely heavily on reference frames combined with masked-frame inpainting [33, 51, 11, 10]. This methodology extracts appearance information from reference frames to inpaint masked regions in target frames—a process that introduces several critical limitations. These methods struggle with head pose variations, identity preservation, and artifact elimination, especially when target poses differ significantly from references [30, 29].

Furthermore, the dependence on explicit masks cannot fully prevent unwanted lip shape leakage, compromising synchronization quality and restricting applicability across diverse visual representations [1]. The challenges intensify in the context of audio-driven generation. Unlike strong visual cues, audio signals provide relatively weak conditioning, making precise lip synchronization difficult [40]. Additionally, existing methods rely on face detection and alignment [3] techniques that break down when applied to stylized characters and non-human entities, precisely the diverse content that modern text-to-video models excel at generating.

This technical gap is compounded by the absence of standardized evaluation frameworks for lip sync in stylized videos. Current benchmarks [52, 42] focus almost exclusively on photorealistic human faces in controlled settings, failing to capture the visual diversity inherent in AI-generated videos.

To address these challenges, we introduce OmniSync, a universal lip synchronization framework designed for diverse videos. Our approach eliminates reliance on reference frames and explicit masks through a diffusion-based direct video editing paradigm. In addition, we establish AIGC-LipSync Benchmark, the first comprehensive evaluation framework for lip synchronization across diverse AIGC contexts. OmniSync's technical approach is built upon three key innovations:

First, we implement a mask-free training paradigm using Diffusion Transformers (DiT) [26] for direct cross-frame editing. Our model learns a mapping function $(V_{cd}, A_{ab}) \mapsto V_{ab}$, where $V$ represents video frames and $A$ represents audio. The indices $(a : b, c : d)$ represent different segments sampled from the same video. The model modifies only speech-relevant regions according to target audio without requiring explicit masks or references. This approach enables unlimited-duration inference while maintaining natural facial dynamics and preserving character identity.

Second, we introduce a flow-matching-based progressive noise initialization strategy during inference. Rather than beginning with random noise [38], we inject controlled noise into original frames using Flow Matching [19], then execute only the final denoising steps. This approach maintains spatial consistency between source and generated frames while allowing sufficient flexibility for precise mouth region modifications, effectively mitigating pose inconsistency and identity drift.

Third, we develop a dynamic spatiotemporal Classifier-Free Guidance (CFG) framework [13] that provides fine-grained control over the generation process. By adaptively adjusting guidance strength across both temporal and spatial dimensions: temporally reducing guidance strength as denoising progresses, and spatially applying Gaussian-weighted control centered on mouth-relevant regions. This balanced approach ensures precise lip synchronization without disturbing unrelated areas.

Our contributions can be summarized as follows:

- A universal lip synchronization framework that eliminates reliance on reference frames and explicit masks, enabling accurate speech synchronization across diverse visual representations.
- A flow-matching-based progressive noise initialization strategy during inference, effectively stabilizing the early denoising process and mitigating pose inconsistency and identity drift.

- A dynamic spatiotemporal CFG framework that provides fine-grained control over audio influence, addressing the weak signal problem in audio-driven generation.

- A comprehensive AIGC-LipSync Benchmark for evaluating lip synchronization in AI-generated content, including stylized characters and non-human entities.

## 2 Related Work

### 2.1 Audio-driven Lip Synchronization

**GAN-based Lip Synchronization.** Traditional GAN-based [9] methods [33, 41, 7, 23, 12] have established important foundations in lip synchronization. Wav2Lip [33] pioneered the use of pretrained SyncNet to supervise generator training, setting a benchmark for subsequent research. DINet [51] enhanced synchronization quality by performing spatial deformation on reference image feature maps, better preserving high-frequency details. IP-LAP [54] introduced a two-stage approach that first infers landmarks from audio before rendering them into facial images. ReSyncer [10] incorporated 3D mesh priors for facial motion, effectively reducing artifacts.

**Diffusion-based Lip Synchronization.** Recent advances in diffusion models [25, 50, 17, 22] have enabled significant progress in audio-driven lip synchronization. LatentSync [17] represents an end-to-end framework based on audio-conditioned latent diffusion models without intermediate motion representation. SayAnything [22] employs a denoising UNet architecture that processes video latents with audio conditioning. MuseTalk [50] proposes a novel sampling strategy that selects reference images with head poses closely matching the target.

However, these methods still rely on reference frames combined with masked-frame inpainting, leading to head pose limitations, identity preservation issues, and blurry edge generation. Our OmniSync framework addresses these limitations through a mask-free training paradigm that enables application across diverse visual representations.

### 2.2 Audio-driven Portrait Animation

Audio-driven portrait animation [38, 46, 15, 6, 14, 27, 37, 49] differs fundamentally from lip sync. Portrait animation [45, 8] follows an image-to-video framework without constraints on head poses or facial expressions, eliminating the need to integrate generated content back into original video. This approach is unsuitable for post-generation lip synchronization in video generation pipelines. In contrast, lip synchronization [33, 17] operates within a video-to-video framework, modifying only lip movements while maintaining compatibility with existing footage. This represents a more constrained task, requiring precise modification of lip regions while preserving surrounding facial features.

Recent models like OmniHuman-1 [18] and Mocha [44] use audio directly as a conditioning signal for image-to-video or text-to-video frameworks. However, due to limitations in talking head datasets, their generative capabilities don't match the versatility of advanced video generation models. This gap highlights why specialized lip synchronization for AI-generated videos remains critical.

## 3 Method

### 3.1 Overview

In this section, we present OmniSync, a universal lip synchronization framework designed for diverse visual content (Fig. 2). Our approach comprises three key components: 1) a mask-free training paradigm that eliminates dependency on reference frames and explicit masks, 2) a flow-matching-based progressive noise initialization strategy for enhanced inference stability, and 3) dynamic spatiotemporal Classifier-Free Guidance (CFG) that optimizes lip sync while preserving facial details. The following subsections provide comprehensive explanations of each component.

### 3.2 Mask-Free Training Paradigm

Traditional lip synchronization methods [33, 50] rely on masked-frame inpainting, isolating the mouth region before generating content based on audio input. Despite their prevalence, these approaches

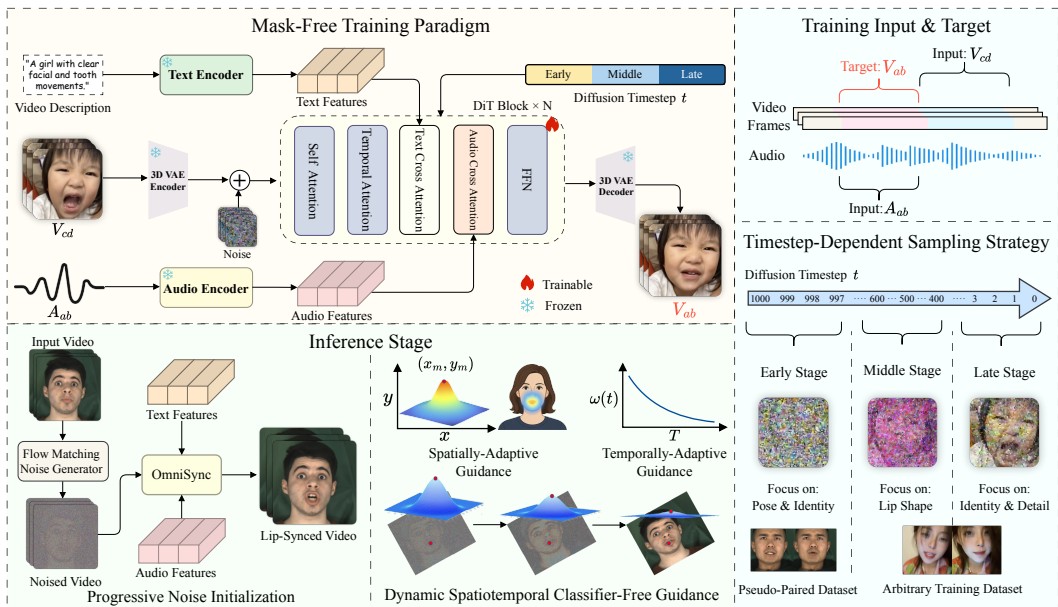

Figure 2: **Overview of OmniSync.** A mask-free training paradigm employs timestep-dependent sampling to predict the lip-synchronized targets $V_{ab}$. During inference, progressive noise initialization and dynamic spatiotemporal CFG ensure consistent head pose and precise lip synchronization.

produce boundary artifacts and struggle with identity preservation. Crucially, they require explicit face detection and alignment—techniques that fail with stylized characters and non-human entities.

An alternative approach is direct frame editing, which aims to transform frames according to target audio without relying on masks or references. However, this approach requires perfectly paired training data with identical head poses and identity—differing only in lip movements. Such paired data is extremely rare and would severely restrict the model's generalizability to diverse visual results.

To address these limitations, we leverage the progressive denoising characteristic of diffusion models, introducing a novel training strategy that varies data sampling based on diffusion timesteps. This allows for stable learning without requiring perfectly paired examples. Our goal is to learn a conditional generation process mapping $(V_{cd}, A_{ab}) \mapsto V_{ab}$ through iterative denoising, where $V$ represents video frames and $A$ represents audio.

We employ Flow Matching [19] as our training objective. Given an input video segment $V_{cd}$ from frames $c$ to $d$, and a target audio segment $A_{ab}$ from frames $a$ to $b$, our model generates the corresponding video segment $V_{ab}$ via the diffusion process:

$$x_{t-1} = \text{DiT}(x_t, V_{cd}, A_{ab}, t), \tag{1}$$

where $x_t$ represents the noised version of target video $V_{ab}$ at timestep $t$, and DiT denotes our diffusion transformer, which predicts the denoised state at timestep $t - 1$.

The CFM loss used for training is defined as:

$$\mathcal{L}_{CFM}(\theta) = \mathbb{E}_{t, x_t, V_{cd}, A_{ab}, V_{ab}} \left[ \| v_\theta(x_t, V_{cd}, A_{ab}, t) - u_t(x_t | V_{ab}) \|_2^2 \right], \tag{2}$$

where $v_\theta(x_t, V_{cd}, A_{ab}, t)$ is the learned velocity field predicted by DiT with conditioning on input video $V_{cd}$ and target audio $A_{ab}$, $u_t(x_t | V_{ab})$ is the conditional velocity field typically defined as $u_t(x_t | V_{ab}) = (V_{ab} - x_t)/(1 - t)$ for the linear interpolation path $x_t = (1 - t)x_0 + tV_{ab}$.

**Timestep-Dependent Sampling Strategy.** A critical insight in our approach is recognizing that the diffusion process can be decomposed into distinct phases, each with different learning requirements. Specifically, early timesteps focus on generating fundamental facial structure, including pose and identity information; middle timesteps primarily generate lip movements driven by audio; while late timesteps refine identity details and textures. To capitalize on this natural progression, we utilize different datasets for distinct timesteps.

For early timesteps (approximately $t \approx T$), responsible for generating overall facial structure, we employ pseudo-paired data from controlled laboratory settings. These samples maintain nearly identical pose information, with variations only in lip movements, providing stable learning signals for structural features and ensuring pose alignment between input and output.

For middle and late timesteps, we transition to more diverse data, sampling from arbitrary videos. During middle timesteps, the model learns to generate lip shapes guided by audio input, whereas in late timesteps (approximately $t \approx 0$), it focuses on refining identity details and ensuring texture consistency. This timestep-dependent training strategy can be formalized as:

$$p(V_{cd}, V_{ab}|t) = \begin{cases} p_{\text{pseudo-paired}}(V_{cd}, V_{ab}) & \text{if } t > t_{\text{threshold}}, \\ p_{\text{arbitrary}}(V_{cd}, V_{ab}) & \text{otherwise.} \end{cases} \tag{3}$$

Here, $p_{\text{pseudo-paired}}$ indicates sampling from controlled datasets with minimal pose variations, while $p_{\text{arbitrary}}$ signifies sampling from our diverse collection of videos. The conditional generation process can be expressed mathematically as:

$$p_\theta(V_{ab}|V_{cd}, A_{ab}) = \int p_\theta(V_{ab}|x_0) p_\theta(x_0|V_{cd}, A_{ab}) dx_0, \tag{4}$$

where $p_\theta(V_{ab}|x_0)$ represents the mapping from the fully denoised state to the output video, and $p_\theta(x_0|V_{cd}, A_{ab})$ captures the relationship between input conditions and the denoised state. Here, $x_0$ refers to the completely denoised latent representation (at timestep $t = 0$).

This progressive training approach aligns well with the natural learning progression of diffusion models. By providing appropriate training signals at each stage, we enable stable learning even without perfectly paired data, allowing our model to generalize effectively to diverse real-world scenarios while maintaining identity consistency.

### 3.3 Progressive Noise Initialization

Standard diffusion-based generation [38] typically begins from random noise (timestep $T$) and progressively denoises toward the final output (timestep 0). However, this approach often results in subtle but noticeable pose misalignments between generated content and original video frames, creating undesirable boundary artifacts and compromising identity preservation.

The fundamental issue lies in error accumulation during the early stages of diffusion. Even minor deviations in early timesteps—when basic facial structure is being formed—can lead to significant misalignments in the final output. This problem is relevant for lip synchronization, where the goal is to modify only speech-relevant regions while maintaining perfect spatial consistency elsewhere. To address this challenge, we introduce a flow-matching-based progressive noise initialization strategy that transforms the traditional diffusion process.

**Flow-Matching Noise Initialization.** Rather than starting the diffusion process from random noise at timestep $T$, we initialize from original video frames with a controlled level of noise. This simulates an intermediate state in the diffusion trajectory, corresponding to a normalized parameter $\tau$. The initialization is performed by adding this controlled noise to the original video frame:

$$x_{\text{init}} = \text{FM}_{\text{add}}(V_{\text{source}}, \tau) = (1 - \tau)V_{\text{source}} + \tau\epsilon, \tag{5}$$

where $x_{\text{init}}$ is the initial noised state derived using the parameter $\tau$, $V_{\text{source}}$ is the source video frame, and $\epsilon \sim \mathcal{N}(0, I)$ is random noise. Let $t_{\text{start}}$ be the discrete timestep corresponding to this initialization point ($T$ is the total number of diffusion steps, and $\tau \in [0, 1]$).

This initialization strategy provides two significant advantages. First, it bypasses the early stages of diffusion (from $T$ down to $t_{\text{start}}$) where general facial structure is formed. This ensures that head pose and global structure are directly inherited from the source frame. Second, it reduces computational requirements by performing denoising only for the remaining steps, from $t_{\text{start}}$ down to 0.

The complete progressive denoising process can be expressed as:

$$x_t = \begin{cases} x_{\text{init}} & \text{if } t = t_{\text{start}}, \\ \text{DiT}(x_{t+1}, V_{\text{source}}, A_{\text{target}}, t + 1) & \text{if } t_{\text{start}} > t \geq 0, \end{cases} \tag{6}$$

where $A_{\text{target}}$ is the target audio used to guide the denoising process, and $t$ here represents discrete diffusion timesteps.

This approach effectively creates a two-stage process: (1) initialization using the flow-matching-inspired noise addition (Eq. 5) to reach a state equivalent to timestep $t_{\text{start}}$, and (2) guided denoising from $t_{\text{start}}$ to 0 that focuses on modifying mouth regions according to the target audio while preserving the overall facial structure, identity features, and head pose from the source frame. By skipping the early noisy stages where basic structures form, we maintain spatial consistency while allowing sufficient flexibility for precise mouth region modifications.

### 3.4 Dynamic Spatiotemporal Classifier-Free Guidance

Audio-driven lip synchronization faces a fundamental challenge: audio signals provide relatively weak conditioning compared to visual cues [40]. Standard Classifier-Free Guidance (CFG) [13] can enhance audio conditioning, but applying uniform guidance across spatial and temporal dimensions creates an unavoidable dilemma: higher guidance scales produce more accurate lip sync but introduce texture artifacts, while lower scales preserve visual fidelity but yield less precise lip movements.

To resolve this tension, we introduce Dynamic Spatiotemporal Classifier-Free Guidance (DS-CFG), a novel approach that provides fine-grained control over the generation process across both spatial and temporal dimensions. Our method applies varying guidance strengths to different regions of the frame and different stages of the diffusion process, achieving an optimal balance between lip synchronization accuracy and overall visual quality.

**Spatially-Adaptive Guidance.** The key insight for spatial adaptation is that audio information primarily affects the mouth region, while other facial areas should remain largely unchanged. We implement this through a Gaussian-weighted spatial guidance matrix that concentrates guidance strength around speech-relevant regions:

$$\mathbf{G}_{\text{spatial}}(x, y) = \omega_{\text{base}} + (\omega_{\text{peak}} - \omega_{\text{base}}) \cdot \exp\left(-\frac{(x - x_m)^2 + (y - y_m)^2}{2\sigma^2}\right) \quad (7)$$

where $(x_m, y_m)$ represents the mouth center, $\sigma$ controls the spread of the Gaussian distribution, $\omega_{\text{base}}$ is the baseline guidance strength applied to non-mouth regions, and $\omega_{\text{peak}}$ is the peak strength applied at the mouth center. This spatial adaptation ensures that audio conditions strongly influence lip and surrounding regions while minimally affecting other facial features.

**Temporally-Adaptive Guidance.** We observe that audio conditioning plays different roles at different stages of the diffusion process. In early diffusion timesteps, strong guidance helps establish correct lip shapes, while in later stages, excessive guidance can disrupt fine texture details. [2, 43, 35] To address this, we implement a temporally decreasing guidance schedule:

$$\omega(t) = \omega_{\text{peak}} \cdot \left(\frac{t}{T}\right)^{\gamma} \quad (8)$$

where $t$ is the current diffusion timestep, $T$ is the total number of timesteps, $\omega_{\text{peak}}$ is the maximum guidance scale, and $\gamma$ controls the decay rate, with a value of 1.5. This temporal adaptation ensures strong guidance during early and middle diffusion stages when coarse structures are formed, gradually reducing influence during later stages when fine details and textures are refined.

**Unified Dynamic Spatiotemporal CFG.** Combining both spatial and temporal adaptations, our DS-CFG approach modifies the standard CFG formulation to:

$$\hat{\epsilon}_{\theta}(x_t, c, t) = \epsilon_{\theta}(x_t, \emptyset, t) + \mathbf{G}_{\text{spatial}} \cdot \omega(t) \cdot (\epsilon_{\theta}(x_t, c, t) - \epsilon_{\theta}(x_t, \emptyset, t)) \quad (9)$$

where $\epsilon_{\theta}(x_t, c, t)$ and $\epsilon_{\theta}(x_t, \emptyset, t)$ are the noise predictions with and without conditioning, respectively.

Through this DS-CFG, our method achieves precise control over the generation process, effectively addressing the weak audio signal problem in audio-driven generation.

## 4 Experiments

### 4.1 Experimental Settings

**Datasets.** We trained OmniSync using the MEAD dataset [42] and a 400-hour dataset collected from YouTube. MEAD's controlled laboratory recordings with diverse facial expressions but minimal

Table 1: Quantitative comparison with previous methods on HDTF Dataset.

| | | HDTF Dataset | | | | | | |
|---|---|---|---|---|---|---|---|---|
| **Method** | **Full Reference Metrics** | | | **No Reference Metrics** | | | **Lip Sync** | |
| | FID ↓ | FVD ↓ | CSIM ↑ | NIQE ↓ | BRISQUE ↓ | HyperIQA ↑ | LMD ↓ | LSE-C ↑ |
| Wav2Lip [33] | 14.912 | 543.340 | 0.852 | 6.495 | 53.372 | 45.822 | 10.007 | 7.630 |
| VideoReTalking [7] | 11.868 | 379.518 | 0.786 | 6.333 | 50.722 | 48.476 | 8.848 | 7.180 |
| TalkLip [41] | 16.680 | 691.518 | 0.843 | 6.377 | 52.109 | 44.393 | 15.954 | 5.880 |
| IP-LAP [54] | 9.512 | 325.691 | 0.809 | 6.533 | 54.402 | 50.086 | 7.695 | 7.260 |
| Diff2Lip [25] | 12.079 | 461.341 | 0.869 | 6.261 | 49.361 | 48.869 | 18.986 | 7.140 |
| MuseTalk [50] | 8.759 | 231.418 | 0.862 | 5.824 | 46.003 | 55.397 | 8.701 | 6.890 |
| LatentSync [17] | 8.518 | 216.899 | 0.859 | 6.270 | 50.861 | 53.208 | 17.344 | **8.050** |
| **Ours** | **7.855** | **199.627** | **0.875** | **5.481** | **37.917** | **56.356** | **7.097** | 7.309 |

Table 2: Quantitative comparison with previous methods on AIGC-LipSync Benchmark.

| | | AIGC-LipSync Benchmark | | | | | | |
|---|---|---|---|---|---|---|---|---|
| **Method** | **Full Reference Metrics** | | | **No Reference Metrics** | | | **Generation Success Rate** | |
| | FID ↓ | FVD ↓ | CSIM ↑ | NIQE ↓ | BRISQUE ↓ | HyperIQA ↑ | All Videos ↑ | Stylized Characters ↑ |
| Wav2Lip [33] | 22.989 | 562.245 | 0.727 | 5.392 | 42.816 | 50.511 | 71.38% | 26.67% |
| VideoReTalking [7] | 20.439 | 329.460 | 0.669 | 5.947 | 45.047 | 48.645 | 48.78% | 7.78% |
| TalkLip [41] | 31.180 | 619.179 | 0.754 | 5.239 | 41.692 | 50.608 | 52.36% | 34.44% |
| IP-LAP [54] | 14.686 | 247.402 | 0.796 | 5.546 | 45.153 | 53.174 | 45.53% | 6.67% |
| Diff2Lip [25] | 23.542 | 403.149 | 0.692 | 5.440 | 42.442 | 50.335 | 74.63% | 36.67% |
| MuseTalk [50] | 17.668 | 297.621 | 0.667 | 4.935 | 36.017 | 58.334 | 92.20% | 67.78% |
| LatentSync [17] | 15.374 | 263.111 | 0.751 | 5.342 | 41.917 | 54.648 | 74.96% | 35.56% |
| **Ours** | **10.681** | **211.350** | **0.808** | **4.588** | **25.485** | **61.906** | **97.40%** | **87.78%** |

head movement provided ideal data for training early denoising stages, while the YouTube dataset enhanced generalization across varied real-world conditions for middle and late stages.

To address the limitations of existing benchmarks that focus solely on real-world videos with frontal views and stable lighting, we created the AIGC-LipSync Benchmark. This comprehensive evaluation framework comprises 615 human-centric videos generated by state-of-the-art text-to-video models such as Kling, Dreamina, Wan [39], and Hunyuan [16]. The benchmark specifically captures challenging visual scenarios such as large facial movements, profile views, variable lighting, occlusions, and stylized characters—conditions that traditional benchmarks fail to address. Details about benchmark construction can be found in the supplementary materials.

**Implementation Details.** We implement our OmniSync model using the Diffusion Transformer architecture. The model is trained on a combined dataset for 80,000 steps using AdamW optimizer [21] with a learning rate of 1e-5. Training is completed in 80 hours using 64 NVIDIA A100 GPUs with a batch size of 64. Audio features are extracted via a pre-trained Whisper encoder, and text conditioning utilizes a T5 encoder. Training employs the timestep-dependent sampling threshold $t_{\text{threshold}} = 850$. The experimental results indicate that excessive thresholds induce significant misalignment while insufficient values will leak the original lip shape. During inference we adopt our flow-matching-based progressive noise initialization starting at $\tau = 0.92$, followed by 50 denoising steps.

## 4.2 Quantitative Evaluation

We evaluate OmniSync against state-of-the-art methods including Wav2Lip [33], VideoReTalking [7], TalkLip [41], IP-LAP [54], Diff2Lip [25], MuseTalk [50], and LatentSync [17] using a comprehensive suite of metrics. For visual quality assessment, we employ FID (Fréchet Inception Distance) to measure frame-level fidelity, FVD (Fréchet Video Distance) for temporal consistency, and CSIM (Cosine Similarity) to quantify identity preservation. Perceptual quality is assessed using no-reference metrics including NIQE (Natural Image Quality Evaluator), BRISQUE (Blind/Referenceless Image Spatial Quality Evaluator), and HyperIQA [36]. For audio-visual synchronization, we measure LMD (Landmark Distance) between predicted and ground truth facial landmarks in the mouth region, and LSE-C (Lip Sync Error - Confidence) to evaluate lip synchronization quality.

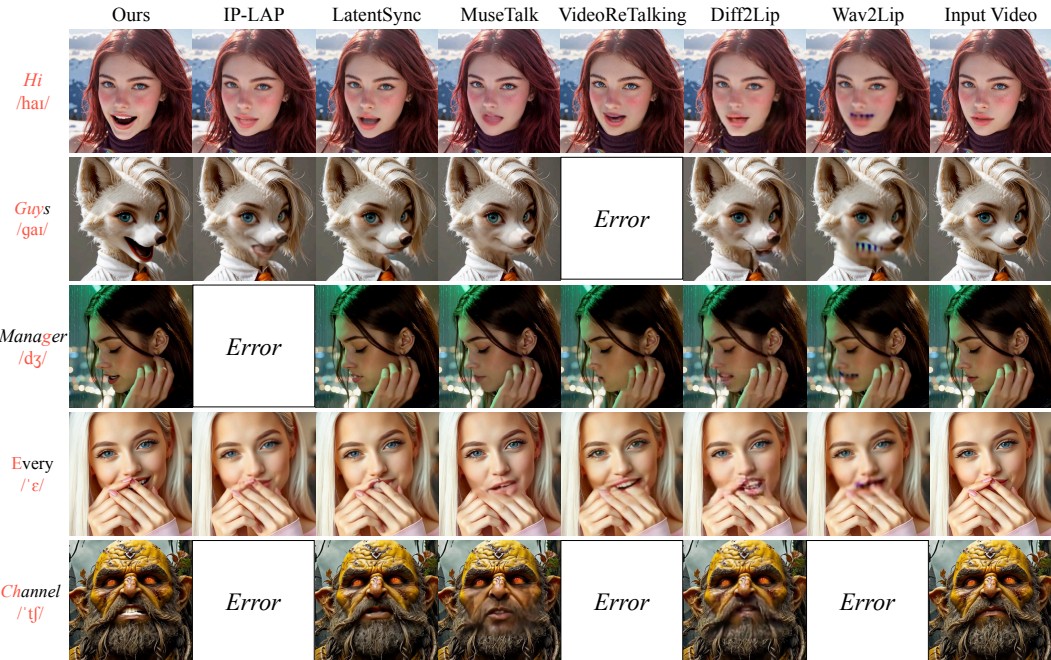

Figure 3: **Qualitative comparison** with previous methods across diverse subjects and phonemes. Our approach produces more accurate lip synchronization and better identity preservation.

Table 3: **User study** results comparing various audio-driven lip-sync methods.

| Metric | Wav2Lip | Video ReTalking | TalkLip | IP-LAP | Diff2Lip | MuseTalk | Latent Sync | Ours |
|---|---|---|---|---|---|---|---|---|
| Lip Sync Accuracy | 2.684 | 2.769 | 1.940 | 2.359 | 3.000 | 2.632 | 3.812 | **3.923** |
| Identity Preservation | 2.410 | 2.786 | 2.222 | 2.991 | 2.889 | 3.197 | 3.658 | **4.128** |
| Timing Stability | 2.556 | 2.376 | 2.162 | 3.034 | 2.812 | 3.145 | 3.581 | **4.043** |
| Image Quality | 2.017 | 2.607 | 1.889 | 3.171 | 2.402 | 3.094 | 3.632 | **4.051** |
| Video Realism | 2.120 | 2.419 | 1.838 | 2.316 | 2.479 | 2.761 | 3.453 | **3.872** |

For the AIGC-LipSync benchmark, we report the Generation Success Rate across all 615 videos and specifically for stylized characters. This metric shows the percentage of videos that are successfully synchronized and pass human verification. This evaluation is essential for universal lip synchronization in AI-generated content, where traditional metrics may not fully capture the challenges of stylized characters, extreme poses, and other atypical visual conditions.

The experimental results in Tab. 1 and Tab. 2 demonstrate that our approach achieves superior performance on multiple metrics. On the HDTF dataset, our method reduced FID by 7.8% and FVD by 8.0% compared to LatentSync, with a remarkable 23.2% improvement in BRISQUE over Diff2Lip. For lip synchronization, we achieved the lowest LMD, outperforming IP-LAP by 7.8%, while LatentSync maintained a slight edge in LSE-C due to its SyncNet-based loss constraint.

On the challenging AIGC-LipSync benchmark, OmniSync demonstrated exceptional capabilities with a 30.5% FID reduction and 19.7% FVD reduction compared to LatentSync, alongside improved identity preservation. Most significantly, our method achieved a 97.40% Generation Success Rate across all videos—substantially higher than MuseTalk (92.20%) and other methods (below 75%). For stylized characters, our success rate of 87.78% outperformed MuseTalk (67.78%), demonstrating OmniSync's capability to handle diverse visual representations including stylized characters.

## 4.3 Qualitative Evaluation

We present qualitative comparisons between OmniSync and existing methods in Fig. 3. Our approach produces more natural facial expressions and superior lip synchronization. Due to lip shape leakage, IP-LAP [54] and LatentSync [17] frequently fail at mouth shape modification, resulting in poor lip synchronization effects. MuseTalk [50] and VideoReTalking [7] modify lip movements but

Table 4: **Ablation study** for our method.

| Methods | FID ↓ | FVD ↓ | CSIM ↑ | NIQE ↓ | BRISQUE ↓ | HyperIQA ↑ | LSE-C ↑ |
|---|---|---|---|---|---|---|---|
| Ours | **15.710** | **287.168** | **0.814** | **5.321** | **29.588** | **57.288** | 7.06 |
| w/o Timestep-Dependent Sampling Strategy | 21.552 | 549.768 | 0.727 | 5.462 | 30.346 | 56.204 | 7.00 |
| w/o Progressive Noise Initialization | 16.731 | 361.282 | 0.805 | 5.349 | 29.789 | 56.511 | 7.03 |
| w/ Low Static CFG | - | - | - | 5.359 | 29.724 | 56.568 | 4.16 |
| w/ High Static CFG | 22.725 | 348.335 | 0.782 | 5.473 | 29.678 | 56.289 | **7.10** |

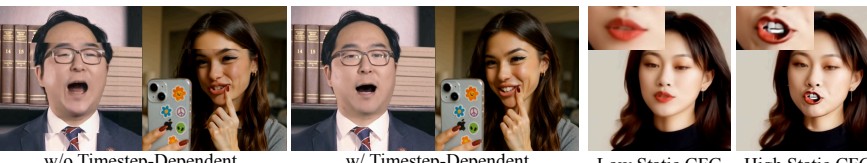

w/o Timestep-Dependent Sampling Strategy     w/ Timestep-Dependent Sampling Strategy     Low Static CFG   High Static CFG   Our DS-CFG

Figure 4: **Ablation study** for timestep-dependent sampling strategy and different CFG settings.

frequently lose identity and visual quality. Diff2Lip [25] and Wav2Lip [33] commonly exhibit lip sync errors, mouth artifacts, and identity drift, particularly in challenging or stylized cases. In contrast, OmniSync consistently maintains identity details and generates realistic, expressive lip movements, demonstrating robust performance. Our approach effectively balances audio and visual cues, addressing the challenge of weak audio conditioning.

**User Study.** To assess perceptual quality, we conducted a user study with 39 participants evaluating 32 video sets generated by OmniSync and seven competing methods, with a standardized Cronbach's $\alpha$ coefficient of 0.98. Participants rated each video on a 5-point Likert scale across five criteria: Lip Sync Accuracy, Character Identity preservation, Timing Stability, Image Quality, and Video Realism. As shown in Tab. 3, OmniSync outperformed all competitors across all metrics, achieving superior scores in Lip Sync Accuracy (3.923 vs. 3.812 for LatentSync), Character Identity (4.128 vs. 3.658), Timing Stability (4.043 vs. 3.581), Image Quality (4.051 vs. 3.632), and Video Realism (3.872 vs. 3.453). These results confirm OmniSync's superior ability to generate high-quality talking videos.

### 4.4 Ablation Study

To clarify the contributions of each core component in our framework, we conduct an ablation study targeting three key modules: the timestep-dependent data sampling strategy, progressive noise initialization, and the Dynamic Spatiotemporal Classifier-Free Guidance (DS-CFG) mechanism. Quantitative results are presented in Tab. 4, and corresponding visual examples are shown in Fig. 4.

Removing the timestep-dependent sampling strategy results in a significant drop in identity preservation and pose consistency, with a 10.7% decrease in CSIM and substantial increases in FID and FVD. As shown in Fig. 4, without this sampling strategy, the generated faces often exhibit clear mismatches with the original image, including noticeable facial misalignment issues. This validates our design choice of aligning pseudo-paired data with early diffusion steps, which proves critical for generating structurally stable outputs. Similarly, removing progressive noise initialization leads to evident temporal inconsistencies and an increase in FVD, confirming the importance of our flow-matching initialization in preserving spatial anchoring and motion coherence.

We also compare our proposed DS-CFG with both low and high static CFG settings. As illustrated in Fig. 4, low CFG provides insufficient audio conditioning, resulting in under-articulated lip movements (LSE-C: 4.16), whereas high CFG improves synchronization (LSE-C: 7.10) but introduces noticeable artifacts and distortions in facial details. In contrast, DS-CFG achieves an optimal balance by applying strong localized guidance in early diffusion stages and gradually reducing it in later steps. These results confirm that dynamic control across temporal and spatial dimensions is essential for producing expressive and visually coherent lip synchronization in generative video content.

# 5 Conclusion

In this paper, we introduce OmniSync, a universal lip synchronization framework for diverse content that addresses critical limitations of traditional approaches. Our three key innovations—a mask-free training paradigm eliminating mask dependencies, a flow-matching-based progressive noise initialization strategy ensuring identity preservation, and dynamic spatiotemporal Classifier-Free Guidance balancing synchronization with visual quality—collectively enable precise lip movements across diverse visual representations. To support systematic evaluation in this field, we establish the AIGC-LipSync Benchmark, the first comprehensive framework for assessing lip synchronization in varied AIGC contexts. Extensive experiments demonstrate OmniSync's superior performance across challenging scenarios, establishing a robust foundation for integrating precise lip synchronization into the broader AI video generation ecosystem.

## Acknowledgments

This work was supported by National Natural Science Foundation of China (NSFC) under Grant Nos. 62436010, 72572090, 62172421 and 62572474. This work was also supported by the Outstanding Innovative Talents Cultivation Funded Programs 2023 of Renmin University of China.

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
