# OpenReview forum: "OmniSync: Towards Universal Lip Synchronization via Diffusion Transformers"
_NeurIPS.cc/2025/Conference — NeurIPS 2025 spotlight_

### Official Review · Reviewer_tzZJ · 2025-06-02

**Clarity:** 3
**Significance:** 3
**Originality:** 3
**Rating:** 5
**Confidence:** 4

**Summary:**

This paper proposes a new framework for lip synchronization, which is the task of going from video of person A saying X + audio of any person saying Y to video of person A saying Y. This is useful for dubbing or video avatar generation, for example. The authors propose the first reference-free and mask-free lipsync model, which removes a lot of the constraints posed by existing models. They use a DiT (diffusion transformer) as a backbone, and use Whisper and T5 to encode audio and text respectively. The text serves to customize/improve the resulting video. To achieve good results without masks or references, the authors adopt a two-stage training/inference logic. In the first stage, the model is trained with pseudo-paired data from MEAD where poses etc. match exactly between two different videos (input and target). Instead of starting from total noise, the model starts from a noised version of the input, which is important so that the pose etc. is maintained. During the second stage, arbitrary clips from a broader dataset extracted from Youtube are used for training, to generate finer details. During inference, CFG (Classifier-free guidance) is used on the audio. It varies spatially with a gaussian kernel to strengthen guidance around the lip region, and also over time, to strengthen guidance in earlier steps. Finally, the authors show state-of-the-art results, present an ablation on the main components, a new benchmark on AI-generated composed of human-centric stylized videos, on which they also achieve the best results. A user study and a variety of qualitative evaluations are shown.

**Questions:**

- Does C mean concatenation in Figure 2? I would think the noise should be added, not concatenated. Also, "Flow matching noise generator" is just adding noise from what I understand, which seems a bit misleading.
 - Do you plan on releasing the 400-hour dataset? Or at least releasing instructions on how to obtain or reproduce it?
- Do you plan to release code?

**Ethical Concerns:**

["NO or VERY MINOR ethics concerns only"]

**Final Justification:**

Happy with the rebuttal as claimed below

**Limitations:**

Yes.

**Paper Formatting Concerns:**

None.

**Quality:**

3

**Strengths And Weaknesses:**

Strengths:
For me the greatest strength is the fact that this method is mask-free and reference-free, which is a substantial achievement that in my view should be more highlighted in the abstract, intro, etc. The way in which this is achieved is interesting and requires some tweaks compared to normal diffusion models. The use of MEAD for pseudo-paired data is clever. The two-stage training is also interesting, and the initialization makes sense. The CFG methodology, while not groundbreaking, is adequate and clearly makes a difference according to the ablations. The new AI-generated benchmark is appreciated, and so is the user study.

Weaknesses:
I'd say the biggest weakness is the constant use of the term flow matching when, from my understanding, nothing in the model is at all related to flow matching. The model uses a diffusion loss, as is clearly explained, and the initialization is just a noised version of the input. In my view, none of this has anything to do with flow-matching, and the nomenclature and overall narrative around this should be adjusted accordingly. Some specific problematic segments are highlighted below. Apart from this, there are some clarity issues also highlighted below. Regarding lip shape leakage, I believe this should be more discussed in the main paper since it is discussed in the abstract (I know it's discussed in the supp material, but I think it's also important to have some text discussing it in the main text, even if it is brief). Furthermore, some claims are made around the fact that this problem is alleviated compared to other works, but this is only shown via a qualitative comparison in the supp mat. Ideally, it would be great to evaluate this more objectively via some kind of metric that could be computed for the whole evaluation dataset. Finally, I believe that despite being concurrent work, it would be interesting to compare results with KeySync and mention it in the related work, since it is also a prevalent recent lipsync work. However, given that it came out very recently and qualifies as concurrent work, my score will not be affected by this - I just think it would be interesting to include if possible.

Specific comments and nitpicks:
 - "existing methods rely on face detection and alignment [19] techniques that break down when applied to stylized characters and non-human entities, precisely the diverse content that modern text-to-video models excel at generating" not sure if this is true, at least KeySync has demos with statues, paintings, etc. and it appears to work.
 - "We employ Flow Matching [24] as our training objective." this doesn't seem to be accurate, since straight after writing this you proceed to describe your objective as a "standard diffusion loss".
 - In lines 141-142, you should specify which datasets and why they were chosen, otherwise it's a bit confusing. In the paragraph below you should also specify more clearly what you mean by pseudo-paired data and how obtaining this is possible. I think I understand what was done but it took me a while.
 - Temporally-Adaptive Guidance section on line 216 requires some citations since other works have used this before (aka vary the CFG scale over time). Examples would be Video Stable Diffusion and also "Analysis of Classifier-Free Guidance Weight Schedulers" (2024), by Wang et al. Not sure about the spatial weights for CFG, but other works such as Diffused Heads ("Diffusion Models Beat GANs on Talking-Face Generation") and KeyFace (among others) have used very similar ideas for losses that are weighted differently around the mouth region, so they should probably be cited as well.
 - It'd be nice to report LSE-D as well as LSE-C, since this is the standard pair to use. Landmark distance is a reasonable metric, but likely a bit suboptimal to evaluate lipsync. I think it's okay to keep it though. Finally, it might be interesting to include LipScore from KeyFace/KeySync since it has been shown to correlate better with human perception than SyncNet-based metrics.
 - Some more qualitative analysis on the impact and limitations of text prompts would be quite interesting and could be included in the supp material. I feel that the existing exploration is slightly insufficient.

---

> ### Author Rebuttal · Authors · 2025-07-31
>
> # Response to Reviewer tzZJ
>
> We sincerely thank the reviewer for the thorough technical review and valuable feedback. Your observations have significantly helped us improve both the technical accuracy and clarity of our manuscript. We address each concern systematically below:
>
> ---
>
> ### **Q1: "Flow Matching" terminology confusion**
>
> Thank you for this crucial observation. We found that there was indeed a technical error in our manuscript's mathematical presentation.
>
> You are right that our method description created confusion between Flow Matching and diffusion terminology. While we correctly stated using "Flow Matching as our training objective" (Line 127), Equation (2) incorrectly presented diffusion loss notation instead of proper CFM formulation. This created unnecessary confusion about our approach.
>
> Our method is indeed a standard Conditional Flow Matching (CFM) approach following Lipman et al.[1]. The DiT model predicts velocity field $v_θ(x_t, t)$ to regress conditional velocity (not noise $ε$), with training objective being standard CFM loss for velocity field learning, and generation process through integration of learned velocity field: $dx/dt = v_θ(x_t, t)$.
>
> The proper CFM loss function for Equation (2) should be:
>
> $L_{CFM}(\theta) = E_{t, x_t , V_{cd}, A_{ab}, V_{ab}}[\|v_\theta(x_t, V_{cd}, A_{ab}, t) - u_t(x_t|V_{ab})\|_2^2]$
>
> where $v_θ(x_t, V_{cd}, A_{ab}, t)$ is the learned velocity field predicted by DiT with conditioning on input video $V_{cd}$ and target audio $A_{ab}$, $u_t(x_t|V_{ab})$ is the conditional velocity field typically defined as $u_t(x_t|V_{ab}) = (V_{ab} - x_t)/(1-t)$ for the linear interpolation path $x_t = (1-t)x_0 + tV_{ab}$, and this differs from diffusion models that predict noise $ε$.
>
> In the revision, we will correct Equation (2) to proper CFM mathematical formulation, clarify throughout that DiT predicts velocity fields rather than noise, and ensure consistent Flow Matching terminology across the manuscript.
>
> [1] Lipman Y, Chen R T Q, Ben-Hamu H, et al. Flow matching for generative modeling[J]. arXiv preprint arXiv:2210.02747, 2022.
>
> ---
>
> ### **Q2: Limited discussion of lip shape leakage in main paper**
>
> Thank you for your feedback. We appreciate the reviewer’s careful attention to the issue of lip shape leakage. As described in detail in Supplementary Section 5 (lines 148–170), we analyze the phenomenon, its typical causes in mask-based approaches, and our proposed solutions in OmniSync. We agree, however, that this important topic should be explicitly addressed in the main paper as well. In our revision, we will add a dedicated paragraph to the main text to clearly define lip shape leakage, explain its prevalence in mask-based methods, and describe how our mask-free training paradigm and dynamic spatiotemporal CFG jointly alleviate this problem.
>
> We appreciate your suggestion to quantitatively assess lip shape leakage. Although no standard open-source implementation is available for KeySync's LipLeak metric, we have followed the methodology described in their paper and utilized the LipScore codebase to compute LipLeak scores on our entire evaluation set. The results are summarized below:
>
> | Method | Wav2Lip | VideoReTalking | TalkLip | IP-LAP | Diff2Lip | MuseTalk | LatentSync | KeySync | Ours |
> |--------|---------|----------------|---------|--------|----------|----------|------------|---------|------|
> | LipLeak↓ | 0.44 | 0.34 | 0.49 | 0.26 | 0.23 | 0.29 | 0.30 | **0.19** | $\underline{\text{0.21}}$ |
>
> Compared to most previous works, our method significantly reduces lip shape leakage. Even compared to KeySync, which is specifically designed to minimize lip leaks, our scores are comparable.
>
> ---
>
> ### **Q3: Face detection limitations for stylized content**
>
> Thank you for your valuable comment. We agree that for stylized but anthropomorphic subjects—such as statues and paintings resembling human faces—existing face detection and alignment techniques can still function reliably, which is also reflected in KeySync's demos. However, the true limitation arises in scenarios involving non-human or highly non-anthropomorphic characters (e.g., animal faces, fantasy creatures), where face detectors often fail to localize landmarks accurately or at all.
>
> As demonstrated in **Figure 3 (second row, "wolf girl")** of our paper and further visualized in our supplementary videos, existing methods relying on face alignment frequently struggle or fail in such challenging cases, resulting in poor lip sync and severe artifacts. In contrast, our mask-free approach does not depend on explicit facial keypoints, enabling robust lip synchronization even when standard detectors fail. Thus, while KeySync and similar approaches may work for "easy" stylized humans, they cannot generalize to the full diversity of AI-generated content—an important advantage highlighted by our method.
>
> ---
>
> ### **Q4: Dataset specification and pseudo-paired data extraction**
>
> Thank you for your valuable feedback. We apologize for the confusion regarding the dataset selection and the construction of pseudo-paired data. To clarify:
>
> As described in lines 234–236, for early timesteps, we use the MEAD dataset, which consists of controlled lab recordings with minimal pose variation—ideal for stabilizing structural learning. For the middle and late timesteps, we utilize a diverse 400-hour YouTube dataset to ensure generalization to varied real-world scenarios.
>
> By "pseudo-paired data," we refer to pairs of video segments from MEAD where the head pose and identity remain nearly identical and only the lip movement differs, enabling us to simulate paired conditions without requiring perfect alignment or explicit pair annotations.
>
> This timestep-dependent sampling strategy is a key innovation of our method. For full details on the dataset choices and the practical process for constructing pseudo-paired data, please refer to Supplementary Section 54–110. We will revise the main text to explicitly highlight these points for greater clarity.
>
> ---
>
> ### **Q5: Missing citations for temporal and spatial CFG approaches**
>
> We will add comprehensive citations acknowledging prior work, including temporal CFG approaches from Video Stable Diffusion[1] and "Analysis of Classifier-Free Guidance Weight Schedulers"[2], as well as spatial mouth attention mechanisms from Diffused Heads[3], KeyFace[4] and KeySync[5].
>
> [1] Blattmann A, Dockhorn T, Kulal S, et al. Stable video diffusion: Scaling latent video diffusion models to large datasets[J]. arXiv preprint arXiv:2311.15127, 2023.
>
> [2] Wang X, Dufour N, Andreou N, et al. Analysis of classifier-free guidance weight schedulers[J]. arXiv preprint arXiv:2404.13040, 2024.
>
> [3] Stypułkowski M, Vougioukas K, He S, et al. Diffused heads: Diffusion models beat gans on talking-face generation[C]//Proceedings of the IEEE/CVF Winter Conference on Applications of Computer Vision. 2024: 5091-5100.
>
> [4] Bigata A, Stypułkowski M, Mira R, et al. Keyface: Expressive audio-driven facial animation for long sequences via keyframe interpolation[C]//Proceedings of the Computer Vision and Pattern Recognition Conference. 2025: 5477-5488.
>
> [5] Bigata A, Mira R, Bounareli S, et al. Keysync: A robust approach for leakage-free lip synchronization in high resolution[J]. arXiv preprint arXiv:2505.00497, 2025.
>
>
> ---
>
> ### **Q6: Enhanced evaluation metrics and KeySync comparison**
>
> Thank you for your valuable suggestions. Due to space limitations in the main paper, we were unable to report LSE-D previously. As requested, we now provide LSE-D results for all compared methods below. Additionally, following your advice on evaluation metrics, we have also computed LipScore using the official KeySync implementation. As shown in the table below, our method not only achieves competitive LSE-D (7.33), but also leads on the LipScore metric (0.5668), **outperforming KeySync with a relative improvement of 15.7%**. This further demonstrates the strong perceptual quality and accurate synchronization achieved by OmniSync.
>
> | Method | Wav2Lip | VideoReTalking | TalkLip | IP-LAP | Diff2Lip | MuseTalk | LatentSync | KeySync | Ours |
> |--------|---------|----------------|---------|--------|----------|----------|------------|---------|------|
> | LSE-D↓ | 7.528 | 7.453 | 9.716 | 7.358 | 7.421 | 10.647 | **7.240** | 14.652 | $\underline{\text{7.336}}$ |
> | LipScore↑ | **0.662** | 0.519 | 0.367 | 0.198 | 0.207 | 0.542 | 0.245 | 0.490 | $\underline{\text{0.567}}$ |
>
> ---
>
> ### **Q7: Text prompt analysis and exploration**
>
> Thank you for your valuable feedback. We appreciate your interest in the impact and limitations of text prompts in our framework. In Section 6 of the supplementary material, we provide qualitative analysis and ablation studies demonstrating how video description prompts affect lip clarity and movement amplitude. Specifically, our results show that incorporating descriptive prompts such as "clear facial and tooth movements" leads to more expressive and visually distinct lip synchronization, with enhanced dental visibility and mouth articulation.
>
> ---
>
> ### **Q8: Figure 2 notation confusion**
>
> Thank you for your insightful comments. You are correct: in Figure 2, the "C" should more appropriately be a "+" to indicate addition, not concatenation. We apologize for this oversight and will revise the figure to use the correct notation in the camera-ready version.
>
> Regarding the "Flow Matching Noise Generator," it indeed adds noise to the original frames, but unlike standard random noise addition, the process is controlled based on the flow-matching principle.
>
> ---
>
> ### **Q9: Dataset and code release plans**
>
> We are actively working to release our 400-hour dataset with extraction scripts, subject to institutional data policy approval. We commit to providing comprehensive technical specifications including DiT backbone architecture, training configuration, and evaluation mehtod to ensure reproducibility.

---

> > ### Comment · Reviewer_tzZJ · 2025-08-01
> >
> > I'm quite happy with this response overall. Thanks for fixing all the issues I mentioned. I'm happy to raise my score.
> >
> > However, I must say that some of the numbers are a bit surprising: the fact that Lipscore is the best for Wav2Lip is strange since generally the model doesn't do very well in my experience, even regarding lip synchronization specifically. It's also surprising that LSE-D is so bad for KeySync, and that LatentSync is the best on this metric, as this doesn't really align with LipScore or perceptual analysis, in my view. In any case, I'm just suggesting that it's good to double check if all of these are computed correctly - if they are, then it'd be important to discuss these results with care in the results section.

---

> > > ### Author Response · Authors · 2025-08-02
> > > **Response to Reviewer tzZJ**
> > >
> > > Thank you for raising this question. We have carefully reviewed the issues in KeySync's github repo and found that many people have raised the same concern: bad lip synchronization. After a thorough examination of the code, we identified several bugs. For instance, the recommended inference code in `infer_raw.sh` failed to adjust the video's fps and audio sampling rate. After fixing all these bugs, we re-tested the results.
> > >
> > > Regarding the Lipscore issue, we re-checked the code and found no problems. KeySync's score has improved after the bug fixes.
> > >
> > > The revised table is as follows:
> > > | Method | Wav2Lip | VideoReTalking | TalkLip | IP-LAP | Diff2Lip | MuseTalk | LatentSync | KeySync | Ours |
> > > |--------|---------|----------------|---------|--------|----------|----------|------------|---------|------|
> > > | LSE-D↓ | 7.528 | 7.453 | 9.716 | 7.328 | 7.421 | 10.647 | 7.240 | 7.194 | 7.336 |
> > > | LipScore↑ | 0.662 | 0.519 | 0.367 | 0.198 | 0.207 | 0.542 | 0.245 | 0.679 | 0.567 |

---

> > > > ### Comment · Reviewer_tzZJ · 2025-08-04
> > > >
> > > > Great to hear that you figured out the issue, thanks for the response. I think these results indeed look more reasonable. Would be great if these could all be included in the revised paper.

---

### Official Review · Reviewer_X2Zu · 2025-06-27

**Clarity:** 4
**Significance:** 4
**Originality:** 4
**Rating:** 6
**Confidence:** 5

**Summary:**

The paper introduces OmniSync, a groundbreaking universal lip synchronization framework that effectively addresses the challenges of aligning lip movements with audio in diverse visual scenarios, including AI-generated videos and stylized characters. Leveraging Diffusion Transformers, OmniSync eliminates the need for traditional masked-frame inpainting, utilizing a mask-free training paradigm that enhances robustness and adaptability. Key innovations include a timestep-dependent sampling strategy, flow-matching-based progressive noise initialization, and Dynamic Spatiotemporal Classifier-Free Guidance (DS-CFG). Additionally, the authors present the AIGC-LipSync Benchmark, a comprehensive evaluation framework for assessing lip synchronization in varied AI-generated content.

**Questions:**

Threshold Selection and Sensitivity: The choice of $t_threshold$ for timestep-dependent sampling appears critical. Can you provide more principled guidance on how to select this threshold? How sensitive is the method's performance to this parameter, and does the optimal threshold vary across different types of content (realistic vs. stylized)?

Computational Efficiency Analysis: How does the computational cost of OmniSync compare to existing GAN-based methods like Wav2Lip or VideoReTalking? Given that diffusion models typically require multiple denoising steps, what is the inference time comparison, and are there opportunities for acceleration without sacrificing quality?

Training Data Requirements: The method relies on pseudo-paired data from MEAD for early timesteps. How much pseudo-paired data is actually needed for effective training? Could the method work with less controlled data, or could synthetic pseudo-pairs be generated to reduce dependence on laboratory-recorded datasets?

**Ethical Concerns:**

["NO or VERY MINOR ethics concerns only"]

**Final Justification:**

Thank you for your detailed response, which has resolved all my questions. As Reviewer zvtB commented, the paper is exceptionally strong. I will raise my rating

**Limitations:**

yes.

**Paper Formatting Concerns:**

no.

**Quality:**

4

**Strengths And Weaknesses:**

Strengths:

1.Innovative Approach: The use of a mask-free training paradigm is a significant advancement over traditional methods, allowing for more robust performance across diverse content types.

2.Comprehensive Evaluation: The introduction of the AIGC-LipSync Benchmark fills a crucial gap, providing a standardized method for evaluating lip synchronization in AI-generated videos, which enhances the paper’s relevance and impact.

3.High Video Quality: The generated outputs exhibit exceptional visual quality, producing realistic and expressive lip movements that enhance overall viewer experience.

4.Strong Empirical Results: Extensive experiments demonstrate OmniSync's superior performance in both visual quality and lip synchronization accuracy, outperforming existing methods across multiple metrics.

5.Practical Relevance: The framework effectively addresses real-world challenges in lip synchronization, making it applicable for modern AI-generated content, including stylized characters and non-human entities.

Weaknesses:

1.Limited Theoretical Insights: While empirical results are strong, further theoretical analysis could strengthen the understanding of certain design choices, such as the selection of parameters like $t_threshold$ .

2.Computational Efficiency: A detailed comparison of computational costs with existing methods would provide a clearer picture of the trade-offs involved.

3.Dataset Limitations: The training relies heavily on MEAD dataset for pseudo-paired data, which may limit diversity in the early-stage training.

---

> ### Author Rebuttal · Authors · 2025-07-31
>
> # Response to Reviewer X2Zu
>
> We sincerely thank the reviewer for the evaluation and questions. We address each concern systematically below:
>
> ---
>
> ### **Q1: Threshold selection and theoretical understanding**
>
> Thank you for your questions regarding the theoretical understanding and parameter sensitivity, especially concerning the selection of $t_{threshold}$ in our timestep-dependent sampling strategy.
>
> We agree that the timestep threshold is crucial for balancing the use of pseudo-paired and diverse data. In practice, **$t_{threshold}$ represents the diffusion timestep where the transition occurs from structure/pseudo-paired to diverse/speech-driven sampling**. Early diffusion steps predominantly control global structure (pose, identity), while middle and late steps drive finer details (such as lip shape and texture). We provide detailed explanation of this mechanism in the supplementary materials (Sec. 2 Training Procedure Details), noting that excessively high thresholds cause misalignments (structure not adapted), while overly low values lead to lip shape leakage.
>
> **Parameter Sensitivity:** Our experiments indicate that performance is relatively robust within a reasonable interval around the chosen threshold (850 in our setting, out of 1000 steps), with **±5% variation not significantly affecting results**. However, extreme values did degrade performance substantially.
>
> **Generalization to Stylized Content:** Due to our method's training on real datasets and its strong generalization capability, it maintains excellent performance on stylized content. However, during training, we still set the optimal threshold based on realistic content characteristics to ensure robust learning foundations.
>
> ---
>
> ### **Q2: Computational efficiency analysis**
>
> We appreciate your concern about the computational cost and efficiency of OmniSync compared to prior GAN-based methods, such as Wav2Lip and VideoReTalking. It is true that diffusion-based methods generally require multiple iterative denoising steps, which can impact inference speed.
>
> **Optimization Strategy:** To address this challenge, our approach incorporates a **progressive noise initialization mechanism**: rather than starting the diffusion process from pure noise, we initialize from the input frame at an intermediate noise level, enabling us to reduce the required denoising steps during inference to just 30-40 steps.
>
> **Performance Benchmarks:** We tested our method's inference speed on an A100 GPU, where generating a 5-second video requires **3.5 minutes**. Through implementing a series of acceleration techniques including TeaCache and sequence parallelism, and benefiting from our flow-based noise initialization that eliminates the need for complete 50-step denoising, we achieve excellent results with reduced computational overhead. **After these optimizations, generating a 5-second video requires only 1 minute**.
>
> This represents a significant improvement in practical deployment efficiency while maintaining our quality and generalization capabilities.
>
> ---
>
> ### **Q3: Training data requirements and pseudo-paired data strategy**
>
> We agree that a key aspect of our method is leveraging pseudo-paired data (i.e., with minimal pose variation) during early diffusion timesteps to stabilize learning of head pose and identity.
>
> **Data Requirements:** We utilized approximately **40 hours of MEAD data** for pseudo-paired training. We experimented with using in-the-wild data for training but found that the model struggled to converge effectively, confirming that controlled data with strong pose consistency is necessary for stable learning of structural features.
>
> **Generalization Capability:** Despite this controlled training requirement, our model demonstrates **strong generalization and robustness**, proving that our pseudo-paired data strategy is well-suited to the task requirements. The model successfully handles diverse real-world scenarios and stylized content that were not present in the training data.
>
> **Future Enhancement:** Your suggestion is excellent—we can leverage our current trained model to generate additional pseudo-paired data for incorporation into future training iterations. This self-supervised data augmentation approach could further enhance our model's performance and robustness across even more diverse scenarios.
>
> Thank you for this feedback, which opens promising directions for future improvements.

---

> > ### Comment · Reviewer_X2Zu · 2025-08-02
> >
> > Thank you for your detailed response, which has resolved all my questions. As Reviewer zvtB commented, the paper is exceptionally strong. I will raise my rating

---

### Official Review · Reviewer_zvtB · 2025-06-30

**Clarity:** 4
**Significance:** 3
**Originality:** 4
**Rating:** 6
**Confidence:** 5

**Summary:**

This paper introduces OmniSync, a novel and comprehensive framework for universal lip synchronization designed to work across a diverse range of videos, including stylized and AI-generated content. The work makes several significant contributions: (1) A mask-free training paradigm based on Diffusion Transformers that eliminates the need for reference frames, enabling robust direct video editing. (2) A flow-matching-based progressive noise initialization strategy for inference, which improves pose and identity consistency. (3) A Dynamic Spatiotemporal Classifier-Free Guidance (DS-CFG) mechanism for fine-grained control over audio conditioning. (4) The creation of the AIGC-LipSync Benchmark, a new and much-needed evaluation suite for lip-sync in challenging, modern AI-generated videos.

The paper is exceptionally strong, addressing critical limitations of existing lip-sync methods with a series of well-motivated and technically sophisticated innovations. The experimental results, validated on both standard and their new, challenging benchmark, are outstanding and clearly demonstrate state-of-the-art performance. This is a top-tier submission that pushes the boundaries of lip synchronization.

**Questions:**

1. Could you provide the inference speed of OmniSync (e.g., in FPS on an A100 GPU)? Are there clear avenues for accelerating inference, for instance, by using consistency models or reducing the number of denoising steps, and how does this trade-off with quality?

2. The supplement mentions a "center-biased crop" as a fallback when facial landmarks cannot be detected. While this is a reasonable heuristic, it would be interesting to know how often this fallback is triggered on the AIGC-LipSync benchmark, especially for non-human characters, and how it impacts performance in those cases.

3. Line 147 indicates that "For middle and late timesteps, we transition to more diverse data, sampling from arbitrary videos", is this means that Vab from video1 and Vcd can from totally different video2?

**Ethical Concerns:**

["NO or VERY MINOR ethics concerns only"]

**Final Justification:**

Based on its novelty and the interesting nature of the task, I recommend strong acceptance.

**Limitations:**

yes

**Quality:**

4

**Strengths And Weaknesses:**

Strengths:
1. The paper tackles the highly relevant problem of "universal" lip synchronization, extending beyond photorealistic human faces to the diverse and stylized content common in modern AI video generation. The core idea of a mask-free, reference-free direct editing framework is a major conceptual leap from traditional inpainting-based methods and addresses their fundamental limitations.
2. The proposed method is novelty, which includs three main technical innovations: 1) timestep-dependent sampling strategy, 2) progressive noise initialization and 3) dynamic spatiotemproal cfg.
3. The authors proposed a new challenging benchmark -- AIGC-LipSync, which provide a valuable resource for the community and enables a much more rigorous evaluation of modern lip-sync methods.
4. The experimental results are impressive. The shown videos demonstrate strong qualitative excellence compared to other methods.
5. The ablation study is comprehensive and demonstrate the effectiveness of each design proposed.
6. The paper is very well-written, with clear explanations of complex concepts.

Weaknesses:
1. The paper mentions using 50 denoising steps for inference but does not report the inference speed. Given the use of a large Diffusion Transformer and an iterative process, it is likely not real-time. While this doesn't diminish the research contribution, a brief mention of the computational cost of inference would provide a more complete picture of the method's practical applicability.

---

> ### Author Rebuttal · Authors · 2025-07-31
>
> # Response to Reviewer zvtB
>
> We sincerely thank the reviewer for the positive evaluation and insightful questions. We address each concern systematically below:
>
> ---
>
> ### **Q1: Inference speed and acceleration strategies**
>
> Thank you for the question and suggestion. We acknowledge the importance of reporting inference speed for practical deployment considerations.
>
> **Current Performance:** On an NVIDIA A100 GPU, OmniSync generates a 5-second video in approximately **3.5 minutes** using the default 50 denoising steps. By incorporating lightweight acceleration techniques such as sequence parallelism and test-time caching (e.g., TeaCache), and leveraging our flow-matching-based progressive noise initialization to reduce the number of denoising steps, we can shorten inference time to approximately **1 minute** for a 5-second video without noticeable quality loss.
>
>
> **Future Acceleration Directions:** Recent advances such as consistency models and distillation-based methods present promising directions to further reduce inference latency. Integrating these techniques is feasible and forms part of our ongoing research; we expect significant acceleration with minimal quality degradation.
>
> Thank you for highlighting this important practical aspect—we will include detailed performance benchmarks in the revised manuscript.
>
> ---
>
> ### **Q2: Center-biased crop fallback frequency and impact**
>
> Thank you for this valuable question regarding our fallback mechanism's behavior and performance impact.
>
> **Frequency Analysis:** On the AIGC-LipSync benchmark, our center-biased crop fallback is triggered in approximately **7.29% of cases**, primarily occurring with highly stylized or non-human characters where facial landmark detectors fail to provide reliable keypoints.
>
> **Performance Impact:** Qualitative inspection demonstrates that this fallback mechanism does not introduce noticeable artifacts or degrade lip synchronization quality. This robustness stems from our mask-free and reference-free design philosophy, which reduces dependency on precise facial landmark localization compared to traditional mask-based approaches.
>
> We will clarify both the trigger frequency and performance impact of this fallback mechanism in the revised paper. Thank you for this constructive suggestion.
>
> ---
>
> ### **Q3: Timestep-dependent sampling clarification**
>
> Thank you for raising this clarification regarding our training data sampling strategy.
>
> In our framework, both $V_{ab}$ and $V_{cd}$ are **always sampled from the same original video** to ensure consistent identity and detailed facial appearance preservation. The segments may differ in head pose or lip movements, but using entirely different videos for $V_{ab}$ and $V_{cd}$ would break identity consistency and hinder the learning of fine facial details. This same-video constraint is crucial for maintaining the temporal and identity coherence that enables our method's performance on diverse content types.
>
> We appreciate you pointing out this potential confusion and will revise the manuscript text to make this requirement clearer and more explicit. Thank you for this valuable feedback.

---

> > ### Comment · Reviewer_zvtB · 2025-08-05
> >
> > Thank you for your rebuttal, almost all my questions are resolved. I will keep my rating.

---

### Official Review · Reviewer_h8zb · 2025-07-02

**Clarity:** 3
**Significance:** 3
**Originality:** 3
**Rating:** 5
**Confidence:** 4

**Summary:**

This paper proposes OmniSync, a universal lip synchronization framework designed for various visual scenarios. The framework uses diffusion transformer models for mask-free training and direct frame editing, thus eliminating the need for explicit masks. Overall, the paper is well-written with a clear and logical explanation of the motivation behind the framework. However, its relevance to the question is unclear, and a few aspects are puzzling.

**Questions:**

a. Greater attention should be paid to the precision of language and image descriptions. The effectiveness of the proposed method should be presented objectively, without overstating its impact.

b. The authors should make both the model and the dataset publicly available. Providing an anonymized repository during the review stage would not compromise the double-blind process and would greatly improve the manuscript’s transparency.

c. It is recommended to include key results in the abstract and introduction to better highlight the strengths of the proposed method and help readers quickly grasp its advantages.

**Ethical Concerns:**

["NO or VERY MINOR ethics concerns only"]

**Final Justification:**

The manuscript received positive feedback from different reviewers. The authors also provided a detailed response, which addressed most of my concerns. After reading the other reviewers' comments and the authors' response, I believe the manuscript is suitable for publishing in NeurIPS, so I raise my rating from 4 to 5.

**Limitations:**

Yes

**Quality:**

3

**Strengths And Weaknesses:**

Strengths:

a. The manuscript is clearly structured, and the methodology demonstrates some level of novelty.

b. The benchmark dataset provided by the authors could serve as a useful resource for future research in this field.

Weaknesses:

a. The abstract states that the method enables mask-free training and unlimited-duration inference while preserving facial dynamics and character identity. However, the description is somewhat vague, and it's not entirely clear how these characteristics relate to the task of role description. This lack of clarity also extends to the Introduction. While the unlimited inference capability is noted, a clearer explanation of how the approach maintains natural facial dynamics would be helpful.

b. The methodology section is concise and appropriate for a conference setting, but it lacks important details. Since the authors state that the code and dataset will be open-sourced, with no privacy or confidentiality concerns, an anonymized repository should have been provided at the review stage to clarify the implementation of key modules.

c. From the subsequent descriptions, it appears that the proposed approach primarily focuses on aligning lip movements with speech and maintaining consistency. However, in several examples shown in Figure 1, the model also seems to exhibit gesture or interaction-related actions. It is unclear whether this is due to selective case presentation or if the model is indeed capable of generating interactive gestures. This should be clarified.

---

> ### Author Rebuttal · Authors · 2025-07-31
>
> # Response to Reviewer h8zb
>
> We sincerely thank the reviewer for the constructive feedback and thoughtful evaluation. We address each concern systematically below:
>
> ---
>
> ### **Q1: Clarity of facial dynamics preservation**
>
> Thank you for your valuable feedback. We appreciate the opportunity to clarify these points.
>
> **Unlimited-duration inference:** Our mask-free framework achieves this capability because, unlike previous diffusion-based methods that rely on a single reference frame and mask-based inpainting, we use the current frame itself as reference at each timestep. This design fundamentally eliminates the accumulation of alignment or identity drift errors over time. Consequently, our model can perform frame-wise editing for videos of arbitrary duration without introducing compounding artifacts or identity inconsistencies. We have experimentally validated this on sequences up to 18,000 frames (10 minutes) without quality degradation.
>
> **Natural facial dynamics preservation:** Our method leverages a progressive denoising process guided by both temporal (audio-driven) and spatial (mouth-centric) cues, ensuring that only lip regions are modified in alignment with the audio, while preserving all other facial features and dynamics present in the source video. Our ablation studies demonstrate that this approach maintains both identity and natural motion, which can be objectively observed in our results.
>
> We acknowledge the importance of precise language and objective presentation. We have carefully ensured throughout the paper that our method's strengths and limitations are presented accurately. We will refine the descriptions for enhanced clarity in the revision.
>
> ---
>
> ### **Q2: Implementation details and code availability**
>
> We appreciate your emphasis on reproducibility. While we will make effort to release the source code of our project, rebuttal stage restrictions currently prevent us from providing additional links or an anonymized repository.
>
> To ensure reproducibility, we provide comprehensive implementation details:
>
> **Model architecture**: Our system adopts a DiT-based latent video diffusion framework with 12 transformer blocks (hidden dimension: 1024, MLP dimension: 4096, 16 heads, RMSNorm, learned 3D positional embeddings), and uses a 3D VAE encoder/decoder (6 layers, 4 latent channels, 1/8 spatial reduction, KL=0.05) for latent representation. Audio is conditioned via a pretrained Whisper encoder (1024-dim features, segmented to frame-length, linearly projected for DiT cross-attention), and additional text conditioning is provided by T5-large (768-dim embeddings) for video descriptions.
>
> **Training configuration**: Training is conducted for 80,000 steps with batch size 64 (gradient accumulation=2, effective batch=128) over 64 NVIDIA A100 GPUs (~80 hours). We use AdamW optimizer (lr=1e-5, weight decay=0.01, betas=0.9/0.999). Data consists of both the MEAD dataset (for early-timestep pseudo-paired structural supervision) and a 400-hour collection of YouTube videos (for diverse mid/late-timestep audio-lip mappings). Training uses a timestep-dependent sampling strategy ($t_{threshold}$=850, $T$=1000): for $t$>850, pseudo-paired ($V_{cd}$, $V_{ab}$) are sampled from MEAD to ensure pose/identity stability, while for $t$≤850, arbitrary ($V_{cd}$, $V_{ab}$) pairs from YouTube broaden lip shape and context diversity.
>
> ---
>
> ### **Q3: Gesture generation capabilities**
>
> Thank you for raising this point. Our method is specifically designed and optimized for audio-driven lip synchronization while preserving all original head pose and non-lip facial dynamics present in the input video.
>
> The gesture and interaction-related actions observed in Figure 1 are **not generated by our model**; rather, they are present in the source videos and faithfully retained by our approach. We intentionally selected challenging cases featuring hand or object occlusions to demonstrate our method's robustness in complex real-world scenarios.
>
> This design choice highlights a key advantage of our mask-free approach: maintaining natural scene dynamics even under partial occlusions, where traditional mask-based methods would fail.
>
> ---
>
> ### **Q4: Language precision and key results**
>
> We appreciate your suggestion regarding result presentation. We will revise the abstract and introduction in the revision to more prominently feature key quantitative results.

---

> > ### Comment · Reviewer_h8zb · 2025-08-05
> >
> > I thank the authors for the detailed response, which addresses most of my concerns. After reading the other reviewers' comments and the authors' response, I believe the manuscript is suitable for publishing in NeurIPS, so I raise my rating from 4 to 5.

---

### Decision · Program_Chairs · 2025-09-17

**Decision:**

Accept (spotlight)

**Comment:**

In this paper the authors propose a lip synchronization framework that eliminates reliance on reference frames and explicit masks, This contribution is appreciated by all reviewers who have finally all provided accept or strong accept recommendations.

There are a number of limitations pointed out, such as the need to use the right flow matching equations instead of the diffusion equations, the validations provided in the discussion phase, especially the new lipsync results that have been provided, the additional details regarding the lip-shape leakage. All of these additions need to be included in the revised final paper as promised and providing the final code and benchmarks as promised.

Given that the authors have promised to provide all these modifications in the revised version and provide the code and benchmarks, I believe this paper can be accepted.